# Cross-Dataset Single-Cell Analysis Identifies Temporal Alterations in Cell Populations of Primary Pancreatic Tumor and Liver Metastasis

**DOI:** 10.3390/cancers15082396

**Published:** 2023-04-21

**Authors:** Daowei Yang, Rohan Moniruzzaman, Hua Wang, Huamin Wang, Yang Chen

**Affiliations:** 1Department of Translational Molecular Pathology, The University of Texas MD Anderson Cancer Center, Houston, TX 77030, USA; 2Department of Pathology, The University of Texas MD Anderson Cancer Center, Houston, TX 77030, USA; 3Department of GI Medical Oncology, The University of Texas MD Anderson Cancer Center, Houston, TX 77030, USA

**Keywords:** pancreatic cancer, tumor microenvironment, single-cell RNA-sequencing analysis, liver metastasis, tumor immunology, genetically engineered mouse models

## Abstract

**Simple Summary:**

Pancreatic cancer has a unique desmoplastic tumor microenvironment composed of various cell populations. However, the direct comparison of cell population composition across early-stage primary tumors, late-stage primary tumors, and metastatic tumors of PDAC is still lacking. In this study, we combined and analyzed single-cell RNA-sequencing analysis (scRNA-seq) datasets of transgenic mouse models with autochthonous pancreatic tumors and liver metastases. Our results revealed the unique tumor ecosystem and cell composition of liver metastases in contrast to primary pancreatic tumors. Specifically, liver metastases exhibited different compositions of major cell populations, including cancer cells, cancer-associated fibroblasts, endothelial cells, lymphocytes, myeloid cells, and granulocytes/neutrophils, compared with primary tumors. We also identified several unique markers, including HMGA1, that were upregulated in cancer cell subpopulations of liver metastases. The unique cell subpopulation composition and genetic profile of liver metastases may provide new insights into potential therapeutic targets and diagnostic markers for metastatic pancreatic cancer.

**Abstract:**

Pancreatic ductal adenocarcinoma (PDAC) has a unique tumor microenvironment composed of various cell populations such as cancer cells, cancer-associated fibroblasts (CAFs), immune cells, and endothelial cells. Recently, single-cell RNA-sequencing analysis (scRNA-seq) has systemically revealed the genomic profiles of these cell populations in PDAC. However, the direct comparison of cell population composition and genomic profile between primary tumors (at both early- and late-stage) and metastatic tumors of PDAC is still lacking. In this study, we combined and analyzed recent scRNA-seq datasets of transgenic KPC mouse models with autochthonous PDAC and matched liver metastasis, revealing the unique tumor ecosystem and cell composition of liver metastasis in contrast to primary PDAC. Metastatic PDAC tumors harbor distinct cancer cell subpopulations from primary tumors. Several unique markers, including HMGA1, were identified for metastasis-enriched cancer cell subpopulations. Furthermore, metastatic tumors reveal significantly enriched granulocytic myeloid-derived suppressor cells (G-MDSCs), mature neutrophils, and granulocyte-myeloid progenitors (GMPs). A common GMP population across primary tumors, liver metastases, and healthy bone marrow was identified as the putative cell origin of tumor-associated neutrophils/granulocytes.

## 1. Introduction

Pancreatic ductal adenocarcinoma (PDAC) is a devastating malignant disease that is difficult to diagnose early and refractory to most therapies. Most patients are present with advanced disease or metastatic disease upon diagnosis [1]. A hallmark feature of PDAC lies in its desmoplastic microenvironment, which is composed of many cell populations such as cancer cells, cancer-associated fibroblasts (CAFs), endothelial cells, and various immune cells. Recent studies using single-cell RNA-sequencing analysis (scRNA-seq) have systemically revealed the genomic profiles of these cell populations in PDAC [2,3,4,5,6,7,8,9,10]. In particular, our recent studies utilizing scRNA-seq and multiple transgenic mouse models revealed the phenotypic alterations in certain cell populations that are associated with various tumor stages and other microenvironmental changes [8,9]. However, it remains unclear whether the significantly different microenvironment of liver metastatic niche, in contrast to that of matched primary pancreatic tumor, would lead to a unique composition of various cell types within the tumor ecosystem. Understanding the key components and interactions in the metastasis-specific ecosystem may provide new opportunities for the treatment or early detection of PDAC metastasis. Here, we conducted a series of cross-dataset analyses of scRNA-seq data on PDAC samples to systemically reveal the differences in tumor microenvironment between primary pancreatic tumor and liver metastasis.

## 2. Methods

### 2.1. Single-Cell RNA-Sequencing (sc-RNA-seq) Analysis

Previously deposited datasets were downloaded from the Gene Expression Omnibus (GEO) repository with the following accession numbers: GSE198815, GSE165534, and GSE184360. Specifically, the scRNA-seq data of early-stage and late-stage primary tumors of *LSL-Kras^G12D/+^*;*Trp53^R172H/+^*;*Pdx1-Cre* (KPC) mice were from the GSE198815 dataset, with the definition for early-stage and late-stage tumors previously described in our original study [8]. Specifically, early-stage PDAC samples were defined as those harboring less than 10% pancreatic adenocarcinoma areas. Late-stage PDAC samples were defined as those harboring greater than 50% pancreatic adenocarcinoma areas. Specifically, the scRNA-seq data of late-stage liver metastases of KPC mice were from the GSE165534 dataset [7]. The scRNA-seq data of normal bone marrow of healthy mice (with the same background as the KPC mice) were from the GSE184360 dataset [11]. All the aforementioned scRNA-seq datasets were derived from unfractionated total live cell mixture from primary pancreatic tumors, liver metastases, or normal bone marrow of background-matched mice. The KPC-Early (early-stage primary tumor) group contains 6204 cells from 3 individual early-stage KPC mice. The KPC-Late (late-stage primary tumor) group contains 5878 cells from 3 individual late-stage KPC mice. The KPC-Late-Met (late-stage liver metastasis) group (3406 cells) was derived from the liver metastases from the same three mice of the KPC-Late group. The bone marrow group contains 2241 cells from 2 healthy mice. All samples were processed using the same protocol by the Sequencing and Microarray Facility at MD Anderson Cancer Center (MDACC). Cells were encapsulated using the 10X Genomics’ Chromium controller and Single Cell 3′ Reagent Kits v2. cDNA was synthesized and amplified to construct Illumina sequencing libraries. The libraries were sequenced by Illumina NextSeq 500. The run format was 26 cycles for read 1, 8 cycles for index 1, and 124 cycles for read 2. Library Seurat version 3.5.3 [12], dplyr, and cowplot were installed into the R package (version 4.1.2) to explore QC metrics, filter cells, normalize data, cluster cells, and identify cluster signature genes. To filter out low-quality cells, a threshold was set as a minimum of 200 and a maximum of 7000 genes per cell. Cells with more than 10% of the mitochondrial genome were also removed for further analysis. “RunUMAP” function was used to cluster the cells. Based on the “JackStrawPlot” and “ElbowPlot” functions, the principal components were used for UMAP projection and clustering analysis. The “FindAllMarkers” function was used to identify the signature genes for each cell cluster. The “FindMarkers” function was used to identify the differentially expressed genes for selected cell clusters between different groups. The “DoHeatmap” function was used to show the top signature genes in each cluster. “VlnPlot” and “DotPlot” functions were used to reveal the expression profiles of selected genes of interest across cell clusters. The cancer cell clusters were analyzed by Monocle3 algorithm to deconvolute and reconstitute the single-cell pseudo-time trajectory [13]. The signature gene lists for the cell clusters and their subpopulations are provided in the spreadsheet files as Appendix A.

### 2.2. Mice

*LSL-Kras^G12D/+^*;*Trp53^R172H/+^*;*Pdx1-Cre* (referred to as KPC) mouse strain was previously documented [14]. KPC strain was maintained with the C57BL/6 background. Both female and male mice with KPC genotype were used for experimental mice to develop spontaneous PDAC. All mice were housed under standard housing conditions at MDACC north campus animal facilities, and all animal procedures were reviewed and approved by the MDACC Institutional Animal Care and Use Committee.

### 2.3. Histology and Immunohistochemistry

Collected fresh mouse tissue samples were fixed in 10% formalin and then processed at MDACC Veterinary Pathology Core Facility to be embedded in paraffin. Embedded tissue samples were sectioned at 5 μm thickness. A human pancreas tissue array containing tissue cores of primary pancreatic tumors and metastatic tumors was purchased from TissueArray.Com LLC (Derwood, MD, USA). Sections were incubated with the following reagents: anti-HMGA1 primary antibody (HPA065612, Sigma Aldrich, St. Louis, MO, USA, 1:100), biotin-conjugated goat anti-rabbit (111-035-144, Jackson ImmunoResearch Laboratories, 1:200), Vectastain ABC kit (PK-6200, Vector Laboratories), DAB substrate kit (SK-4100, Vector Laboratories), and hematoxylin counterstain. Images were captured with an Olympus IX51 microscope and an Olympus Microscope Camera with cellSens Dimension Software version 1.18 (Olympus America Inc., Center Valley, PA, USA). The staining intensity of HMGA1 was quantified by visual scoring of staining on a scale of 0–3 (3—high, 2—medium, 1—low, and 0—negative). The IHC scores of HMGA1 for all samples were graded by combined scores of the intensity of staining and the percentage of positive tumor cells. The formula for staining score was used: S = p1 × 1 + p2 × 2 + p3 × 3, in which p1, p2, and p3 represent fractions of tumor cells representing each staining category of 1, 2, and 3, respectively.

### 2.4. Cell Viability Assay

In vitro cell viability assay was conducted as previously described by our recent study [15] with minor modifications. KPC cancer cells (established from pancreatic tumors from KPC mice) growing in 96-well plates (3 × 10^3^ cells per well) were transfected with either non-targeting universal negative control siRNA (Sigma-Aldrich, SIC001) or predesigned siRNA of *Hmga1* (Sigma-Aldrich, SASI_Mm01_00156613). Transfection was performed using the Lipofectamine RNAiMAX Reagent (Invitrogen, Waltham, MA, USA, 13778075) according to manufacturer’s instructions. Cell viability was examined by Cell Counting Kit 8 (Abcam, Cambridge, UK, ab228554) at 48 h after siRNA transfection.

### 2.5. Data Availability

There are no additional unpublished datasets from this study. All datasets analyzed in this study can be accessed at the National Center for Biotechnology Information’s Gene Expression Omnibus (GEO) database repository with the following accession numbers: GSE198815, GSE165534, and GSE184360. The dataset of human pancreatic tumors can be accessed at the Genome Sequence Archive with the following accession number: CRA001160. The survival and gene expression data of TCGA pancreatic adenocarcinoma cohort were based on the GDAC Firehose PAAD dataset (previously known as the TCGA Provisional dataset).

### 2.6. Statistical Analysis

Statistical analyses of immunostaining quantifications were performed with unpaired, two-tailed *t*-test or one-way ANOVA with Tukey’s multiple comparison test using GraphPad Prism version 9.2.0 (GraphPad Software). The expression data of indicated genes among TCGA pancreatic adenocarcinoma patient samples were based on the RNA Seq V2 RSEM data. TCGA data were downloaded from the cBioPortal database [16,17]. For the survival analysis based on *HMGA1* expression, patients with available OS data and RNA-seq data (*n* = 179) were stratified into *HMGA1*-high (*n* = 57) and *HMGA1*-low (*n* = 122) groups based on the average *HMGA1* expression value. Kaplan–Meier plots were drawn for survival analysis and the log rank Mantel–Cox test was used to evaluate statistical differences. Data met the assumptions of each statistical test; where variance was not equal (determined by an *F*-test), Welch’s correction for unequal variances was applied. A *p* value < 0.05 was considered statistically significant. Error bars represented standard error of the mean (S.E.M.) when multiple visual fields were averaged to produce a single value for each animal, which was then averaged again to represent the mean bar for the group in each graph.

## 3. Results

### 3.1. Single-Cell Analysis Reveals Unique Changes in the Composition and Genomic Profile of Cell Populations in the Liver Metastasis of PDAC

We first conducted a cross-dataset analysis of single-cell RNA-sequencing analysis (scRNA-seq) data on the primary tumors and liver metastatic tumors of *LSL-Kras^G12D/+^*;*Trp53^R172H/+^*;*Pdx1-Cre* (KPC) mice [7,8] with matched animal background and scRNA-seq methodology (Figure 1A,B). Primary tumors of early-stage KPC mice (KPC-Early), primary tumors of late-stage KPC mice (KPC-Late), and liver metastases of matched late-stage KPC mice (KPC-Late-Met) were compared (Figure 1C and Appendix A). Matched primary tumors and liver metastases were collected from the same late-stage KPC mice and simultaneously examined by scRNA-seq. The overall composition of major cell populations among KPC-Early, KPC-Late, and KPC-Late-Met exhibited profound differences (Figure 1C,D). Specifically, the composition of cancer cell subpopulations was significantly altered as tumors progressed from early-stage to late-stage and metastatic stage. KPC-Early tumors revealed the dominant presence of the cancer cell 1 subcluster, while KPC-Late tumors contained three distinct subclusters (namely, cancer cell 1, 2, and 3). In contrast, KPC-Late-Met (liver metastasis) tumors exhibited cancer cell 1 and 3 subclusters, but not cancer cell 2 subcluster (Figure 1C,D). Endothelial cells in liver metastases revealed a distinct subcluster from the endothelial cells in early-stage and late-stage primary tumors (Figure 1C,D). Consistent with our previous observations [8], the subcluster distribution of cancer-associated fibroblasts (CAFs) significantly shifted between KPC-Early and KPC-Late. In comparison, KPC-Late-Met revealed a significantly lower abundance of total CAFs (Figure 1C,D). T cell and B cell numbers were relatively high in KPC-Early and profoundly decreased in KPC-Late and KPC-Late-Met (Figure 1C,D). The significant decrease of T and B cells in late-stage primary tumors was accompanied by the prominent accumulation of neutrophils/granulocytes and myeloid cells. In particular, the number of granulocytic myeloid-derived suppressor cells (G-MDSCs) and mature neutrophils were significantly higher in KPC-Late than KPC-Early, and further elevated in KPC-Late-Met (Figure 1C,D). Myeloid cells also revealed different subcluster compositions with enriched myeloid cell 1 subtype in KPC-Early tumors, enriched myeloid cell 2 subtype in KPC-Late tumors, and enriched myeloid cell 3 subtype in KPC-Late-Met tumors (Figure 1C,D).

### 3.2. Liver Metastases Enrich Certain Cancer Cell Subclusters Compared to Primary Tumors

In general, we identified three major cancer cell subclusters (Figure 2A), namely, cancer cell 1 (CC-1), cancer cell 2 (CC-2), and cancer cell 3 (CC-3). All these three cancer cell subtypes expressed the generic pancreatic cancer cell marker genes, such as *Krt8* and *Krt18* (Figure 2B). Specifically, CC-1 represented the epithelial-like cancer cell subpopulation expressing *Epcam*, *Cdh1*, and *Krt7* (Figure 2B,C). CC-2 subcluster expressed low expression levels of *Epcam*, *Cdh1*, and *Krt7*, while having high expression levels of *Cdkn2a*, *Krt20*, and *Ifitm1* (Figure 2B,C). CC-3 represented the mesenchymal-like cancer cell subpopulation that expressed mesenchymal genes such as *Prrx1*, *Col6a1*, *Thy1*, and *Vim*, but not epithelial marker genes such as *Epcam* and *Cdh1* (Figure 2B,C). Interestingly, CC-3 subcluster highly expressed *Grem1* (Figure 2B,C), which has been shown to regulate pancreatic cancer cell heterogeneity by recent studies [18].

Next, we examined whether the various stages (KPC-Early, KPC-Late, and KPC-Late-Met) of PDAC favor the presence of certain cancer cell subtypes. CC-1 was the predominant subtype in early-stage KPC primary tumors, whereas CC-2 emerged in the late-stage KPC primary tumors (Figure 2D). Interestingly, liver metastases favored the accumulation of CC-1 and CC-3 but not CC-2 (Figure 2D), although the matched late-stage primary tumors consisted of CC-1, CC-2, and CC-3. To correlate our cancer cell subclusters with the previously defined PDAC subtypes [19,20,21], we examined the expression profiles of previously reported signature genes of classical and basal-like subtypes (Figure 2E and Appendix A). In particular, CC-1 exhibited high expression levels of classical subtype genes, such as *Lgals4*, *Agr2*, *Tff2*, and *Vsig2*. CC-2 expressed high levels of basal-like subtype genes, including *Lgals1*, *Fbln2*, and *Areg*. CC-3 expressed high expression levels of not only *Lgals1*, *Fbln2*, and *Areg* (shared with CC-2), but also other mesenchymal genes, such as *Ctsl*, *Sparc*, *Zeb1*, and *Fn1* (Figure 2E). By analyzing the human PDAC scRNA-seq dataset (Genome Sequence Archive: CRA001160) from a recent study [5], we observed similar cancer cell subclusters that resembled the CC-1, CC-2, and CC-3 subtypes in human pancreatic tumors (Appendix A). Human pancreatic tumors also revealed an additional intermediate cancer cell (namely, CC-1/2) subpopulation with a mixed feature between CC-1 and CC-2 subtypes (Appendix A).

### 3.3. Pseudotime Analysis Reveals the Trajectory of Various Cancer Cell Subpopulations

Next, we utilized the Monocle3 pseudotime analysis [13] to reveal the trajectory of various cancer cell subpopulations throughout various stages (KPC-Early, KPC-Late, and KPC-Late-Met) of PDAC (Figure 3A,B). The CC-1 subcluster, dominantly enriched in KPC-Early tumors, was defined as the starting state of the cancer cell trajectory, referred to as ‘Cluster 1’ by the Monocle3 pseudotime analysis (Figure 3A,B). The subsequent trajectory, transitional state, and signature gene expression profile of CC-2 (Monocle3 ‘Cluster 2’) and CC-3 (Monocle3 ‘Clusters 3–9’) were also demonstrated (Figure 3B,C).

To identify the cancer cell-specific signature genes associated with liver metastasis, we analyzed the differentially expressed genes between cancer cells of metastatic tumors and cancer cells of matched late-stage primary tumors (Figure 3D). We identified a variety of genes that exhibit significantly higher expression levels in cancer cells of liver metastases than in cancer cells of primary tumors. Of note, several of these genes (such as *Grem1*, *Hmga1*, and *Krt13*) are specifically expressed by cancer cells, but not by other cell populations (Appendix A), indicating that these genes are potential markers associated with tumor metastasis.

### 3.4. Liver Metastases Reveal Elevated Expression of HMGA1 in Cancer Cells Compared to Primary Tumors

Based on the cancer cell-specific, metastasis-related markers revealed by our prior results (Figure 3D and Appendix A), we further tested the expression of HMGA1 as a proof of concept because of the direct availability of validated antibodies by the Human Protein Atlas database. Cancer cells of metastatic tumors exhibited significantly higher expression level of HMGA1 than cancer cells of primary tumors from matched late-stage KPC mice (Figure 4A,B). In comparison, the expression levels of HMGA1 in early-stage disease areas and adjacent normal areas from KPC mice were much lower (Figure 4A,B).

Consistent results were observed from the HMGA1 staining on human pancreatic tumors (Figure 4C,D). Cancer cells in metastatic tumors exhibited a significantly higher expression level of HMGA1 than those in primary tumors of human patients. Furthermore, a higher expression level of *HMGA1* was associated with worse overall survival and disease-free survival in patients of TCGA pancreatic cancer cohort (Figure 4E).

Next, we conducted in vitro experiments and identified that siRNA-mediated knockdown of *Hmga1* significantly suppressed the cell viability of KPC cancer cells (Appendix A), consistent with recent findings [22]. Taken together, these findings suggest functional contributions of HMGA1 in the progression of PDAC.

### 3.5. Compositions of Endothelial Cells, Fibroblasts, T Cells, and B Cells Vary among Early-Stage, Late-Stage, and Metastatic Tumors

Primary tumors and liver metastases revealed two different subclusters of endothelial cells, referred to as EC-1 and EC-2 subpopulations (Figure 5A). EC-1 subpopulation was the only endothelial cell cluster in the primary tumors, with the expression of *Cd34*, *Cav1*, and *Tie1* (Figure 5B,C). Liver metastases had a very low abundance of EC-1 subtype cells, while revealing a unique endothelial cell subtype (EC-2) noted by the unique expression of *Mrc1* but low expression of *Cd34* and *Cav1* (Figure 5B,C). In addition, the upregulated genes in liver metastasis-specific endothelial cell subcluster included *Tspan7*, *Fcgr2b*, and *Adam23* (Figure 5C). The generic endothelial cell signature genes shared between EC-1 and EC-2 included *Esam*, *Pecam1*, and *Eng* (Figure 5C). The existence of the metastasis-specific EC-2 cluster is presumably associated with the distinct resident endothelial cell population and overall microenvironment in the liver compared with those in primary pancreatic tumors.

Our recent study [8] systemically investigated the impact of different disease stages on the cell subtype compositions of cancer-associated fibroblasts (CAFs) with six various subpopulations, namely, aCAF, bCAF, cCAF, dCAF, eCAF, and fCAF (Figure 5D). In correlation with the CAF subclustering strategy in previous studies [3], aCAF and bCAF resembled the inflammatory CAF (iCAF) subtype; cCAF constituted the myofibroblast (myCAF) subtype; dCAF resembled an intermediate subpopulation with a mixed phenotype between iCAF and myCAF; eCAF and fCAF constituted the antigen-presenting CAF (apCAF) subtype (Appendix A). Here, we further overlayed the single-cell dataset on liver metastases in comparison with our previous datasets on early-stage and late-stage primary tumors (Figure 5E). The liver metastases of late-stage KPC mice rendered less CAFs than matched primary tumors (Figure 5E), despite the exact same methodology used for the simultaneous preparation of scRNA-seq samples from primary tumors and liver metastases. It is unclear whether the different CAF numbers could be due to the different tissue textures between primary tumors and liver metastases that resulted in different yields of CAFs during sample preparation.

We also analyzed the cell composition of T cells and B cells (Figure 6A,B). In general, T cell and B cell subpopulations in liver metastases exhibited similar cell abundance and composition compared to those in matched late-stage primary tumors (Figure 6C,D). The disease progression from early-stage to late-stage was associated with a significant decrease of almost all T cell and B cell subpopulations in the primary tumors (Figure 6C,D). Liver metastases also revealed a low abundance of T and B cells, which was similar to that in late-stage primary tumors. However, a more detailed analysis revealed the subtle increase of certain cell subpopulations in the liver metastases compared to those in matched late-stage primary tumors (Figure 6D). Liver metastases exhibited an increase of a few cell subpopulations, such as NKT cells, CD4 regulatory T cells (Tregs), and CD8^+^ T cells (Figure 6D).

### 3.6. Myeloid Cell Subpopulations Alter among Early-Stage, Late-Stage, and Metastatic Tumors

Next, we compared the composition of myeloid cell populations (namely, myeloid cell 1, 2, and 3) between primary tumors and liver metastases (Figure 7A,B). As shown by our prior results (Figure 1C,D), early-stage primary tumors exhibited a high abundance of myeloid cell 1 subcluster with high expression levels of Major Histocompatibility Complex Class II (MHC-II) genes, such as *H2-Aa* and *H2-Ab1* (Figure 7B–D). In contrast, late-stage primary tumors showed a dominant presence of myeloid cell 2 subcluster with high *Arg1* expression and a low expression of MHC-II genes (Figure 7B–D). Interestingly, liver metastasis exhibited the presence of all three myeloid cell subclusters with similar abundance, which was different from the polarized myeloid cell compositions in early-stage and late-stage primary tumors (Figure 1D and Figure 7B–D). Particularly, liver metastases revealed the most prevalent presence of myeloid cell 3 subcluster (Figure 1D) with high expression levels of *Ifitm6* and *Ly6c2* (Figure 7C,D). In addition, liver metastases (KPC-Late-Met) revealed the enrichment of liver-specific Kupffer cells but lacked conventional dendritic cell 3 (cDC3) subpopulation (Figure 7B).

### 3.7. Cross-Comparison between Primary Tumor, Liver Metastasis, and Normal Bone Marrow Identifies Significantly Enriched Neutrophil/Granulocyte Subtypes in Liver Metastases

The overview of cell population compositions revealed the enrichment of neutrophil/granulocyte populations in late-stage primary tumors and liver metastases, but not early-stage primary tumors (Figure 1D). Specifically, the neutrophil/granulocyte in primary tumors and liver metastases exhibited three major subclusters: granulocyte-myeloid progenitors (GMPs), mature neutrophils (mature-N), and granulocytic myeloid-derived suppressor cells (G-MDSCs). The GMP subcluster, also called neutrophil progenitors [23], exhibited high expression levels of *Elane*, *Mpo*, and *Ctsg* (Appendix A). The mature neutrophil (mature-N) subcluster presented the expression of *Ngp*, *Camp*, *Ltf*, and *Ly6g* (Appendix A). G-MDSC subcluster expressed *Il1b*, *Arg2*, and *Ifitm1* (Appendix A). The mature-N and G-MDSC subclusters were significantly enriched in late-stage primary tumors, more than in early-stage primary tumors (Figure 1D). In contrast, the percentages of GMP, mature-N, and G-MDSC subclusters were significantly elevated in liver metastases compared to those in late-stage primary tumors (Figure 1D). Particularly, G-MDSC subcluster became the most abundant cell subtype among all cell types in liver metastases (Figure 1D).

Next, we combined multiple datasets overlaying normal bone marrow of background-matched tumor-free mice in comparison with KPC primary tumors (both early- and late-stage) and KPC liver metastases (Figure 8A–C). Interestingly, the overlay of these datasets revealed a communal cell population of granulocyte-myeloid progenitors (GMPs) among primary PDAC tumors, liver metastases, and healthy bone marrow. GMPs enriched in the liver metastases (Figure 1D) exhibited analogous transcriptional phenotypes to GMPs from healthy bone marrow (Figure 8D). These results support the possibility that the tumor-associated neutrophils/granulocytes/G-MDSCs differentiate from a common GMP lineage [24] originated from the bone marrow. Further differentiated neutrophils and G-MDSC subclusters in primary tumors and liver metastases revealed apparent divergence from bone-marrow neutrophils (Figure 8D). Specifically, the mature neutrophil subcluster in healthy bone marrow (marrow-N) expressed *Hba-a1*, *Hba-a2*, *Hbb-bs*, and *Hbb-bt*, which were absent in the mature neutrophils of primary tumors and liver metastases.

## 4. Conclusions

Pancreatic cancer develops a desmoplastic tumor microenvironment that is composed of abundant extracellular matrix components and various cell populations. Recent studies utilizing new techniques, such as single-cell sequencing analysis and spatially-resolved sequencing analysis, have revealed the heterogeneity and dynamic alteration of the components of tumor microenvironment at various stages of primary pancreatic tumor progression [2,3,4,5,6,8]. However, the genetic profiles of various cell populations in the liver metastasis have not been systemically investigated in direct comparison with those in primary tumors. Here, we conducted the cross-dataset analyses of single-cell RNA-sequencing (scRNA-seq) data comparing liver metastasis and primary tumors from matched KPC mice, revealing intriguing alterations in the phenotypes and transcriptional profiles of various cell populations. These differences are presumably associated with the distinct microenvironment in the liver metastases compared to that in the primary tumors. Our study depicts the distinct cell population trajectories across early-stage primary tumors, late-stage primary tumors, and late-stage liver metastases.

By analyzing the cancer cell subclusters from liver metastases and primary tumors, we identified the different cancer cell subcluster compositions and gene expression profiles in the liver metastases in contrast to matched primary tumors. Early-stage tumors from KPC mice dominantly exhibited the epithelial-like CC-1 cluster, while late-stage tumors from KPC mice revealed three cancer cell subclusters: CC-1, CC-2, and CC-3. Liver metastases showed the enrichment of CC-1 and CC-3 subclusters, but not CC-2 subcluster. We also observed similar cancer cell subclusters in the scRNA-seq dataset of human pancreatic tumors [5]. Due to the current lack of an available scRNA-seq dataset on liver metastases of human PDAC, it is still unknown whether the cancer cells in liver metastases of human PDAC may have analogous phenotypes to what we observed on KPC mouse models. Further studies and analyses are needed to investigate the landscape alterations of cancer cells, fibroblasts, endothelial cells, and immune cells across primer PDAC tumor, liver metastasis of PDAC, and liver metastasis of other cancer types. In addition, more data are needed to compare the aforementioned cell populations between liver metastasis and normal liver to identify potential metastasis-specific features. Furthermore, we identified that cancer cell subclusters in primary tumors and liver metastases revealed intriguing differences in the signature gene expression profiles of classical and basal-like subtypes [19,20,21]. Pseudotime analysis demonstrated the trajectory correlation between these cancer cell subclusters. Furthermore, liver metastases also exhibited higher expression levels of a variety of cancer cell-associated genes (such as *Hmga1* and *Grem1*) compared to primary tumors. We further validated the upregulated expression level of HMGA1 in cancer cells from liver metastases, in comparison to primary tumors, using immunohistochemistry staining on tissue sections from both transgenic mice and human patients. High expression of HMGA1 was associated with poor survival in TCGA pancreatic cancer patient cohort. Given the recently reported roles of HMGA1 in PDAC progression [22], it is important to further investigate the functional contribution of HMGA1 to the liver metastasis of PDAC.

Our results highlighted the distinct phenotypes and transcriptional profiles of myeloid cell populations in liver metastases compared to those in the primary tumors. Early-stage tumors from KPC mice dominantly exhibited the ‘myeloid cell-1’ subcluster (with high expression levels of MHC-II genes), while late-stage tumors from KPC mice enriched the ‘myeloid cell-2’ subcluster (with high expression levels of *Arg1*). In contrast, liver metastases showed the enrichment of the ‘myeloid cell-3’ subcluster (with high expression levels of *Ly6c2* and *Ifitm6*), while having myeloid cell-1 and myeloid cell-2 subclusters as well.

Last but not the least, liver metastases revealed significant enrichment of neutrophil/granulocyte subpopulations (including the granulocyte-myeloid progenitors, mature neutrophils, and granulocytic-MDSCs) compared to primary tumors. In addition, we cross-compared the scRNA-seq datasets of background-matched healthy mouse bone marrow with primary pancreatic tumors and liver metastases. Furthermore, we identified that the granulocyte-myeloid progenitors (GMPs), enriched in liver metastases, shared a similar transcriptional phenotype to those in healthy bone marrow. These results indicate that the GMPs from the bone marrow are likely the cell source of the neutrophils/granulocytes in primary tumors and liver metastases. The significantly different landscapes of immune cell populations between liver metastasis and primary pancreatic tumors also indicate an intriguing possibility of altered tumor microbiome profiles (for bacterial and fungal species), which merits further investigations.

Taken together, we conducted a comprehensive, cross-dataset analysis on the scRNA-seq data of primary tumors and liver metastases of pancreatic cancer, revealing the distinct phenotypes and transcriptional profiles of various cell populations at different disease stages. Future studies are still needed to identify the mechanisms of liver metastasis through further understanding of the key factors governing the unique tumor microenvironment of liver metastases.

## Figures and Tables

**Figure 1 cancers-15-02396-f001:**
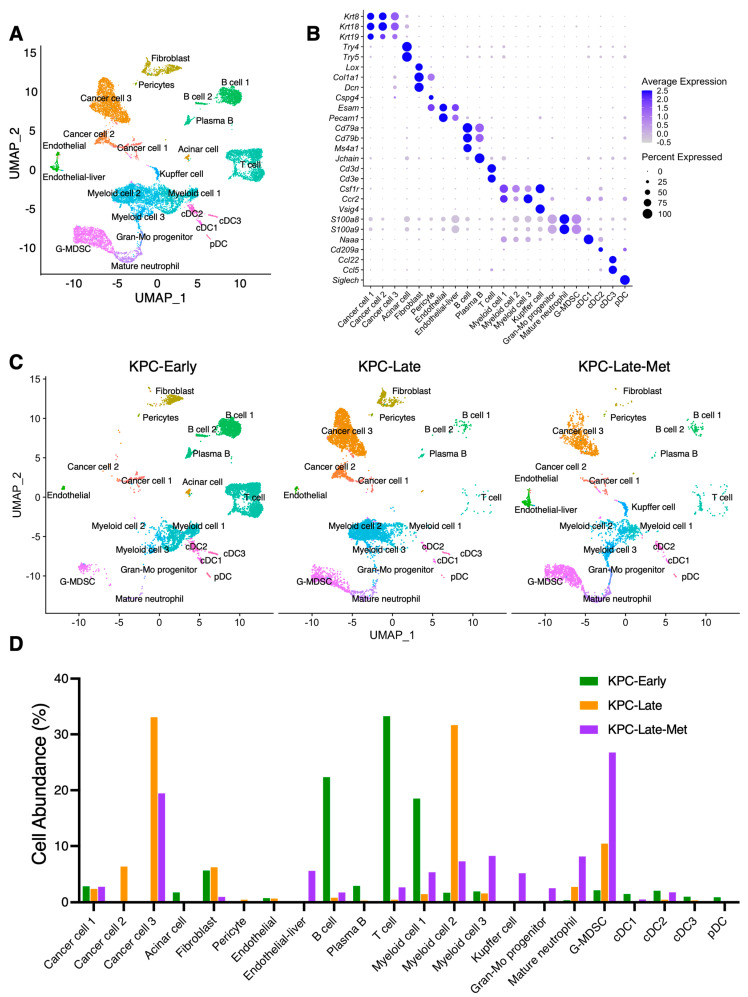
Single-cell RNA-sequencing (scRNA-seq) analysis identifies distinct compositions of cell populations between primary tumors and liver metastases of pancreatic cancer. (**A**,**B**) scRNA-seq analysis of unfractionated live cell mixture from primary pancreatic tumors and liver metastases of *LSL-Kras^G12D/+^*;*Trp53^R172H/+^*;*Pdx1-Cre* (KPC) mice, as published by previous datasets (GSE198815 and GSE165534). The major cell clusters are shown in UMAP plot (**A**). Expression profile of indicated marker genes among defined cell clusters is shown in dot plot with the normalized expression levels of indicated genes (**B**). (**C**) UMAP plot comparison of the distinct cell compositions across three groups of KPC mice (*n* = 3 per group): early-stage primary tumors (KPC-Early), late-stage primary tumors (KPC-Late), and late-stage liver metastatic tumors (KPC-Late-Met). The KPC-Early group contains 6204 cells from 3 individual early-stage KPC mice. The KPC-Late group contains 5878 cells from 3 individual late-stage KPC mice. The KPC-Late-Met group (3406 cells) was derived from the liver metastases from the same three mice of the KPC-Late group. (**D**) The abundance (%) of indicated cell populations across KPC-Early, KPC-Late, and KPC-Late-Met groups. DC, dendritic cell; pDC, plasmacytoid dendritic cell; G-MDSC, granulocytic myeloid-derived suppressor cell; Gran-Mo progenitor, granulocyte-monocyte progenitor cell.

**Figure 2 cancers-15-02396-f002:**
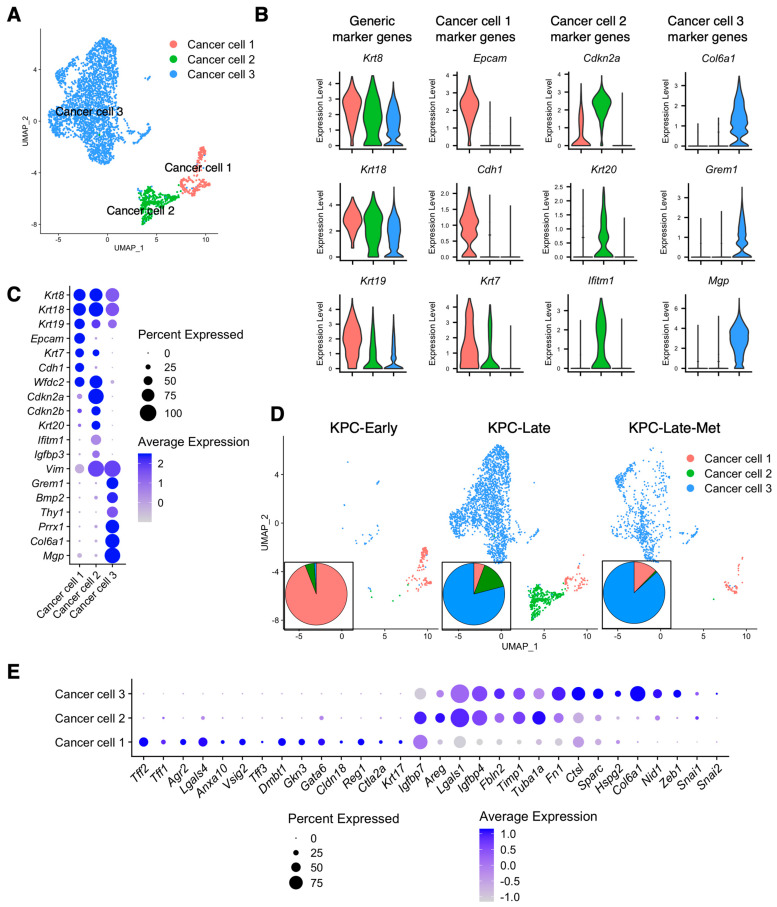
Liver metastases reveal different cancer cell subpopulation compositions from primary tumors. (**A**–**C**) Cancer cells from primary tumors and liver metastases of KPC-Early, KPC-Late, and KPC-Late-Met groups are stratified into three distinct subclusters in UMAP plot (**A**), based on the various expression profiles of signature genes shown in violin plot (**B**) and dot plot (**C**). (**D**) UMAP plot comparison of the distinct cancer cell subcluster compositions in the primary tumors and liver metastases of KPC-Early, KPC-Late, and KPC-Late-Met groups. The temporal alterations in cancer cell subcluster compositions among three groups are also shown in pie chart plot. (**E**) The cancer cell subclusters were examined for the expression profiles of genes associated with the classical and basal-like cancer cell subtype definition, as shown in dot plot.

**Figure 3 cancers-15-02396-f003:**
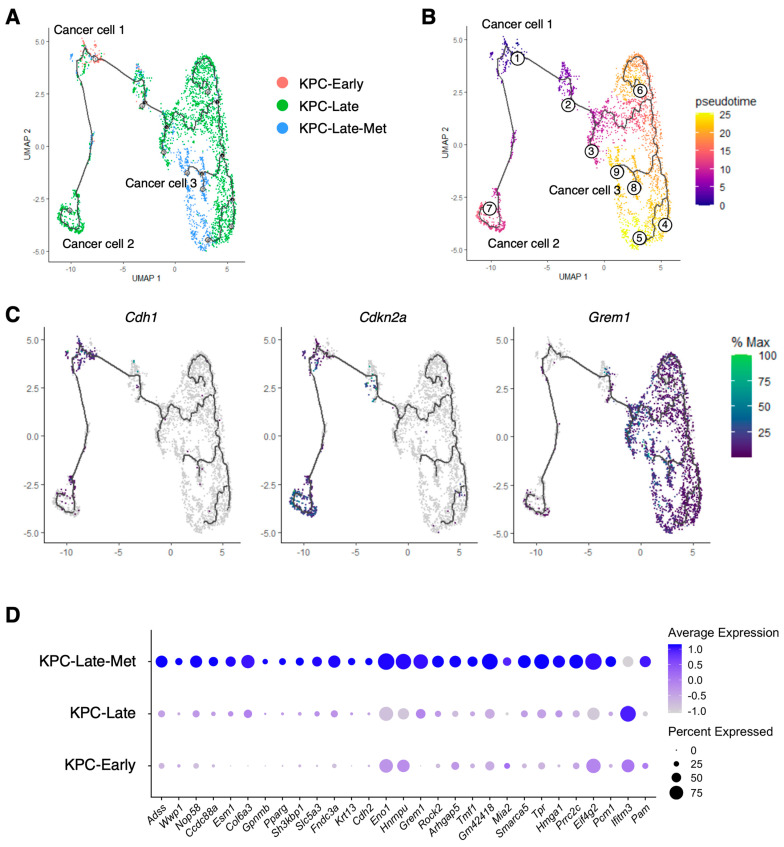
Liver metastases reveal different trajectories of cancer cell subpopulations from primary tumors. (**A**–**C**) Pseudotime trajectory of cancer cell subpopulations, as revealed by the Monocle3 inference analysis. The distributions of cancer cell subpopulations among KPC-Early, KPC-Late, and KPC-Late-Met groups are compared in (**A**). The transitional trajectory of cell subpopulations (defined by Monocle3 as the pseudotime clusters 1–9) is shown in (**B**). Cluster 1 in this Monocle3 trajectory plot represents the previously defined cancer cell CC-1 subcluster as the starting point of the pseudotime trajectory. Cluster 2 represents the previously defined CC-2 subcluster. Clusters 3–9 represent the previously defined CC-3 subcluster. The expression profiles of representative genes for the previously defined CC-1, CC-2, and CC-3 are shown in (**C**). (**D**) Differentially expressed genes (DEGs) were calculated by comparing cancer cells from liver metastasis with cancer cells from matched primary tumors. The metastasis-upregulated genes were defined as the DEGs that were expressed by cancer cells of liver metastases at significantly higher levels than cancer cells of late-stage primary tumors. The expression profiles of indicated metastasis-upregulated genes are shown in dot plot.

**Figure 4 cancers-15-02396-f004:**
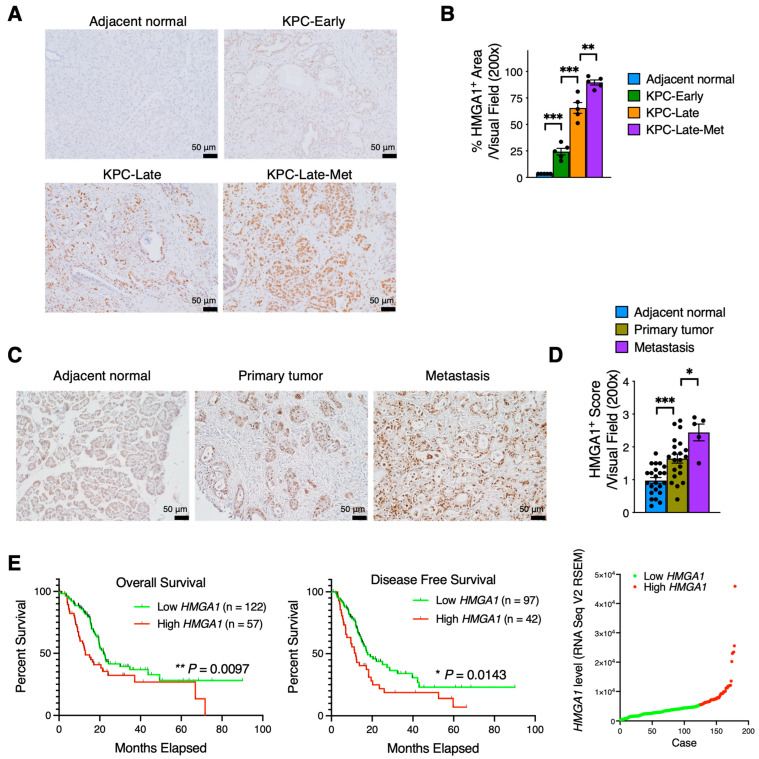
The expression level of HMGA1 is upregulated in cancer cells of liver metastases. (**A**,**B**) HMGA1 immunohistochemistry staining on tumor tissue sections from KPC mice at various stages of PDAC progression (*n* = 5 mice per group). The representative images (**A**) and quantitative results (**B**) of HMGA1 staining are shown. **, *p* < 0.01; ***, *p* < 0.001. (**C**,**D**) HMGA1 immunohistochemistry staining on tumor tissue arrays from human PDAC patients. The representative images (**C**) and quantitative results (**D**) of HMGA1 staining are shown. *, *p* < 0.05; ***, *p* < 0.001. (**E**) Survival of pancreatic adenocarcinoma patients from TCGA dataset correlated with *HMGA1* expression level. Log-rank (Mantel–Cox) test was used. Patients with available survival data and RNA-seq data (*n* = 179) were stratified into two groups based on the average *HMGA1* expression level as the cut-off value. Scatterplot illustrating the distribution of *HMGA1* gene expression levels among PDAC patients is also shown. *, *p* < 0.05; **, *p* < 0.01.

**Figure 5 cancers-15-02396-f005:**
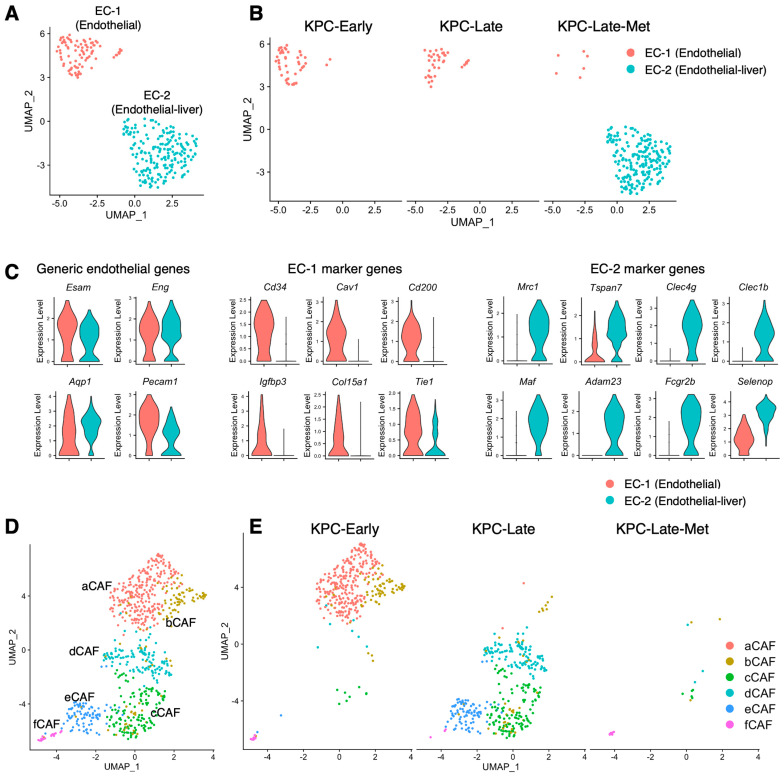
Cell compositions of subclusters of endothelial cells and fibroblasts in primary tumors and liver metastases. (**A**–**C**) Endothelial cells from primary tumors and liver metastases are characterized in UMAP plot (**A**) and compared across KPC-Early, KPC-Late, and KPC-Late-Met groups (**B**). The signature genes of endothelial cells from primary tumors or endothelial cells from liver metastases are shown in violin plot (**C**). (**D**,**E**) Various subpopulations of cancer-associated fibroblasts (CAFs) from primary tumors and liver metastases are characterized in UMAP plot (**D**) and compared across KPC-Early, KPC-Late, and KPC-Late-Met groups^©^.

**Figure 6 cancers-15-02396-f006:**
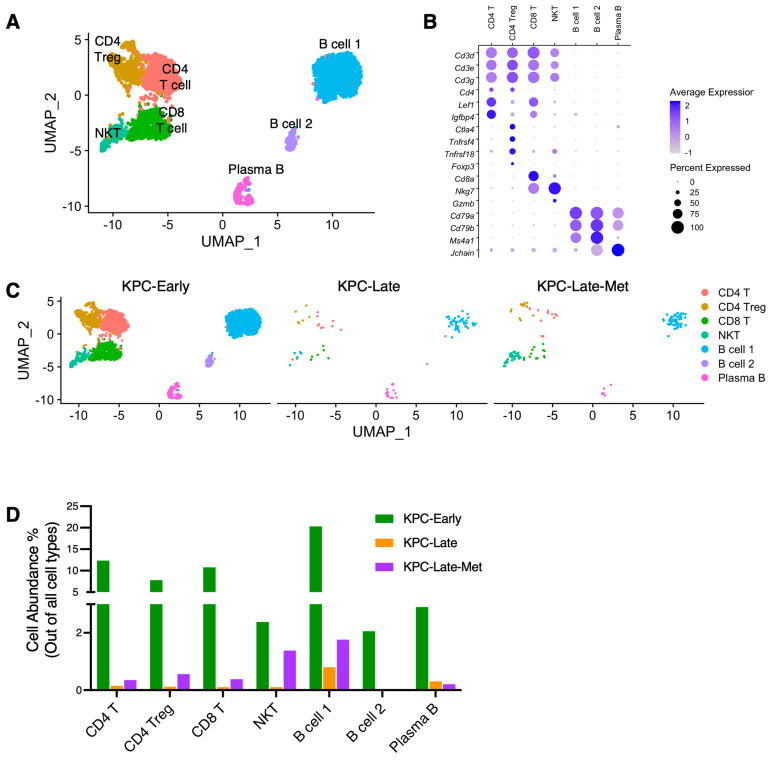
Subpopulation compositions of T cells and B cells in primary tumors and liver metastases. (**A**–**C**) T cells and B cells from primary tumors and liver metastases are stratified into indicated subclusters in UMAP plot (**A**) based on the various expression profiles of signature genes shown in dot plot (**B**). The temporal alteration of T cell and B cell compositions are compared across KPC-Early, KPC-Late, and KPC-Late-Met groups (**C**). (**D**) The abundance (%) of indicated T and B cell subpopulations across KPC-Early, KPC-Late, and KPC-Late-Met groups.

**Figure 7 cancers-15-02396-f007:**
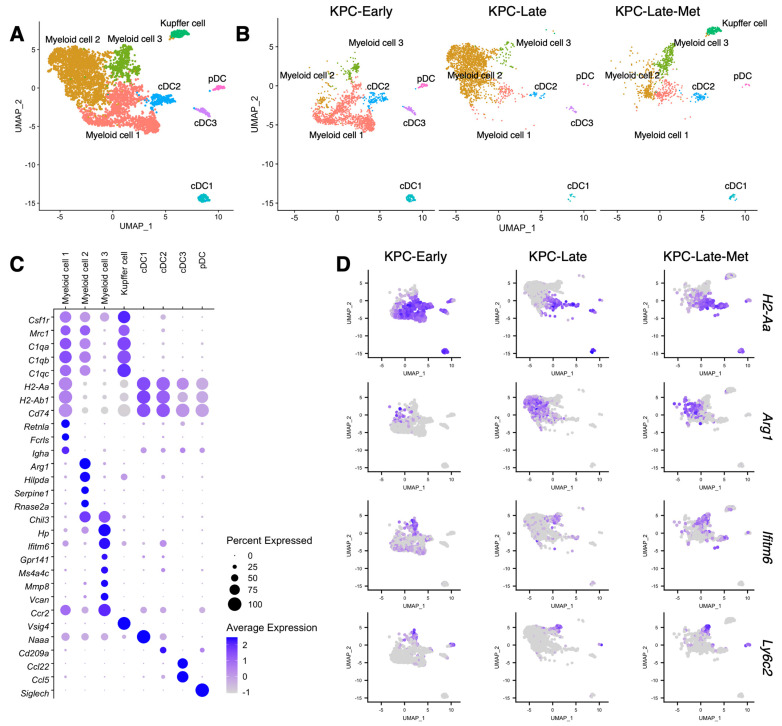
Distinct compositions of myeloid cells and dendritic cells in liver metastases. (**A**–**C**) Myeloid cells and dendritic cells (DCs) from primary tumors and liver metastases are characterized in UMAP plot (**A**) and compared across KPC-Early, KPC-Late, and KPC-Late-Met groups (**B**). The signature genes of myeloid cells and DCs in primary tumors and liver metastases are shown in dot plot (**C**). (**D**) The expression profiles of indicated signature genes of myeloid cell subpopulations in primary tumors and liver metastases are shown in UMAP plot.

**Figure 8 cancers-15-02396-f008:**
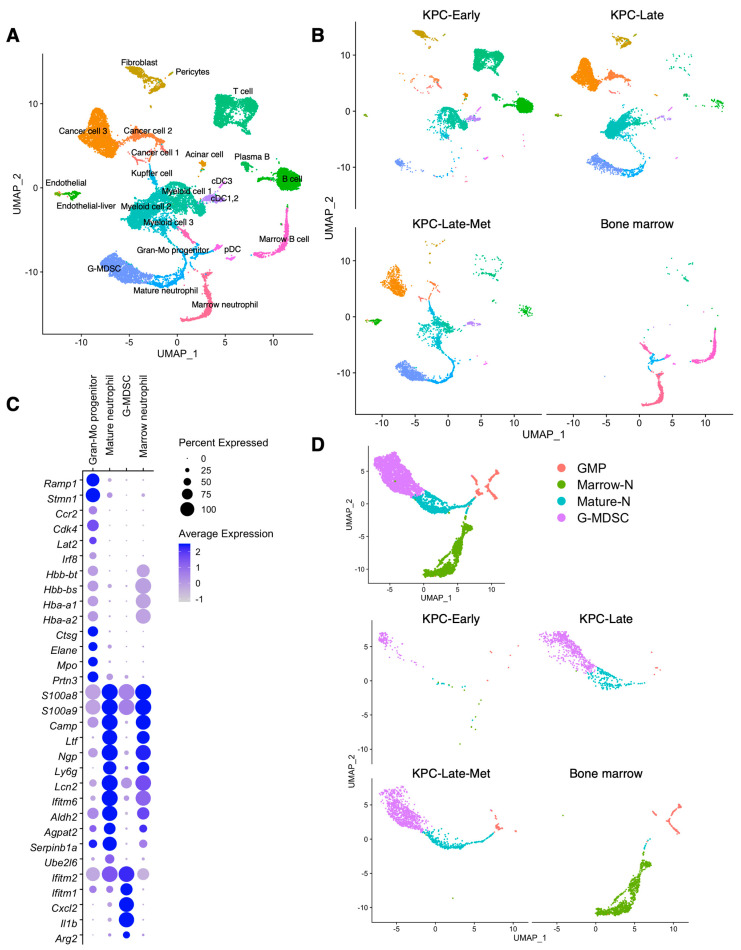
The similarity and difference of granulocyte/neutrophil subclusters across primary tumors, liver metastases, and normal bone marrow. (**A**,**B**) Cross-comparison of scRNA-seq datasets of KPC-Early, KPC-Late, and KPC-Late-Met groups compared with normal bone marrow of background-matched healthy mice (*n* = 2; total cell number 2241), as published by previous datasets (GSE184360). The major cell clusters were defined using the same threshold used in Figure 1, as shown in UMAP plot (**A**). The cell population compositions are compared across KPC-Early, KPC-Late, KPC-Late-Met, and bone marrow groups (**B**). (**C**,**D**) Expression profile of indicated marker genes among defined granulocyte/neutrophil subpopulations is shown in dot plot with the normalized expression levels of indicated signature genes (**C**). Granulocyte/neutrophil subpopulations from primary tumors, liver metastases, and normal bone marrow are characterized in UMAP plot and compared across KPC-Early, KPC-Late, KPC-Late-Met, and bone marrow groups (**D**). G-MDSC, granulocytic myeloid-derived suppressor cell; Gran-Mo progenitor, granulocyte-monocyte progenitor cell.

## Data Availability

All datasets of transgenic mouse models analyzed in this study can be accessed at the National Center for Biotechnology Information’s Gene Expression Omnibus (GEO) database repository with the following accession numbers: GSE198815, GSE165534, and GSE184360. The dataset of human pancreatic tumors can be accessed at the Genome Sequence Archive with the following accession number: CRA001160. The survival and gene expression data of TCGA pancreatic adenocarcinoma cohort were based on the GDAC Firehose PAAD dataset.

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
