# Peer review of "Cross-Dataset Single-Cell Analysis Identifies Temporal Alterations in Cell Populations of Primary Pancreatic Tumor and Liver Metastasis"

_cancers, 2023, doi:10.3390/cancers15082396_

Round 1
Reviewer 1 Report
In this study, the authors performed analyses of scRNA-seq datasets to compare primary PDAC tumors at early and late stages with liver metastasis at late stage. They identify subpopulations of cancer cells, endothelial cells, cancer-associated fibroblasts, T cells, B cells and myeloid cells that differ between different stages of PDAC and liver metastasis. Within the cancer cell populations, they perform pseudotime trajectory and differential expression gene analysis to identify upregulated genes in liver metastasis, proceeding to validate upregulation of one of the genes they find, HMGA1 in murine and human samples. They further analyzed changes of immune subsets between primary early tumor, late tumor and liver metastasis. Overall the study is well designed and executed. Please kindly address the following points to improve the manuscript.
Major points:
1. The analysis in the manuscript is largely derived from murine datasets and lacks further validatory analysis from human datasets. Co-analysis with published human PDAC scRNAseq datasets to find shared patterns will strengthen the study.
2. Compared to the other cell types, the CAF analysis is very brief and somewhat limited. Considering the prominent role of CAF in PDAC, more extensive analysis should be included.
3. The shifts described in liver metastasis would perhaps be more informative if further compared to normal liver.
4. HMGA1 shows interesting association with metastasis based on expression. What about the function of HMGA1 in PDAC? Some experiments to show its function in vitro or in vivo should be included.
Author Response
Response to Reviewer 1
In this study, the authors performed analyses of scRNA-seq datasets to compare primary PDAC tumors at early and late stages with liver metastasis at late stage. They identify subpopulations of cancer cells, endothelial cells, cancer-associated fibroblasts, T cells, B cells and myeloid cells that differ between different stages of PDAC and liver metastasis. Within the cancer cell populations, they perform pseudotime trajectory and differential expression gene analysis to identify upregulated genes in liver metastasis, proceeding to validate upregulation of one of the genes they find, HMGA1 in murine and human samples. They further analyzed changes of immune subsets between primary early tumor, late tumor and liver metastasis. Overall the study is well designed and executed. Please kindly address the following points to improve the manuscript.
Response: We greatly appreciate the reviewer’s overall enthusiasm and insightful comments. As suggested, we have addressed all of the points raised with new data and new experiments, to improve our manuscript. We have also added new discussions according to these comments.
- The analysis in the manuscript is largely derived from murine datasets and lacks further validatory analysis from human datasets. Co-analysis with published human PDAC scRNAseq datasets to find shared patterns will strengthen the study.
Response: We sincerely appreciate this important point. The great majority of cases in PDAC patients are inoperable upon diagnosis, which significantly hinders the sample availability of both primary tumors and liver metastases for single-cell RNA-sequencing (scRNA-seq) analysis. Therefore, the available scRNA-seq dataset for liver metastases from human PDAC patients is still lacking. Furthermore, human PDAC scRNA-seq dataset with matched primary tumors and liver metastases is extremely difficult to obtain, because the majority PDAC of patients (already with locally invasive or metastatic diseases) are inoperable. Therefore, the transgenic mouse models with spontaneous pancreatic tumors and liver metastases (as used in this manuscript and our previous studies) provide indispensable resources allowing for the direct comparison between matched primary tumors and liver metastases.
In order to establish the correlation between our findings on transgenic mouse models and human PDAC, we analyzed the published scRNA-seq dataset of primary pancreatic tumors from human patients (Peng et al., Cell Research, 2019). We indeed observed similar cancer cell (CC-1, CC-2, and CC-3) subtypes (new Supplementary Figure 2C and 2D) based on the signature gene profiles of our definition. In addition, we have added more discussion on this point in the revised Discussion section.
- Compared to the other cell types, the CAF analysis is very brief and somewhat limited. Considering the prominent role of CAF in PDAC, more extensive analysis should be included.
Response: Thank you for this comment. We completely agree with the reviewer on the importance of CAFs in PDAC. This current manuscript analyzed our recently published datasets (GSE198815) by our previous study (McAndrews, Chen, Darpolor et al., Cancer Discovery, 2022), which already thoroughly compared the compositions of CAF subpopulations (a-fCAFs) between early-stage KPC model and late-stage KPC model. Therefore, the previous version of this manuscript did not include extensive amount of analyses on CAF subpopulations. According to this comment, we have conducted more analyses (new Supplementary Figure 4A and 4B) to show the phenotypes and gene signature profiles of CAF subclusters. Specifically, we added new gene expression heatmap plot and dot plot overlaying the myCAF/iCAF/apCAF signature genes on a-fCAF subclusters in the new Supplementary Figure 4A and 4B. Also, we added more discussion on this point.
- The shifts described in liver metastasis would perhaps be more informative if further compared to normal liver.
Response: Thank you for this comment. We agree with the reviewer that phenotypic comparison of cell populations between primary tumor, liver metastasis, and normal liver would be very intriguing and important. Currently, there is limited available scRNA-seq dataset on normal liver samples from published literatures that can be directly compared/overlayed with our scRNA-seq dataset. We are preparing for submitting normal liver samples for scRNA-seq analysis. However, the sequencing process and data analysis (comparing the microenvironment between liver metastasis and normal liver) will require a significant amount of time, which would exceed the time limit and current scope of this current manuscript. We will investigate this important point in our future studies. Nevertheless, we added more discussion on this point in the revised Discussion section.
- HMGA1 shows interesting association with metastasis based on expression. What about the function of HMGA1 in PDAC? Some experiments to show its function in vitro or in vivo should be included.
Response: We appreciate this important point. We completely agree with the reviewer that it would be interesting to further investigate the functions of HMGA1 in PDAC, especially in PDAC metastasis. We have added new data from in vitro functional experiments using pancreatic cancer cells with Hmga1 knockdown. We observed decreased cell viability of pancreatic cancer cells upon siRNA-mediated Hmga1 knockdown (new Supplementary Figure 3B). We have added detailed discussions regarding this point. We are also preparing for injecting mouse pancreatic cancer cells into mice after Hmga1 knockdown, to investigate the impact of Hmga1 knockdown on pancreatic cancer cell metastasis in vivo. However, such in vivo experiments would need a significant amount of time, which would exceed the time limit and current scope of this current revision. According to this comment, we hope to thoroughly investigate the functional contribution of HMGA1 to the liver metastasis of PDAC in our future studies.
Reviewer 2 Report
Overall, this is an excellent study and should be published with no modifications or only minor modifications. The methods and datasets are very well explained. And, there is an SOM available for data transparency.
This reviewer has only three minor requests: An IACUC approval number should be provided, as a matter of the paper trail for anyone interested in verifications; and the following sentence can be changed to “We first conducted a cross-dataset analysis…”; and because of the neutrophil results, addressed at end of Discussion, it would be useful to have some comments about bacteria in the microenvironment.
There is one final issue: While this reviewer did have access to two supplementary figures, despite repeated requests to the editorial office, this reviewer did not have access to the supplementary tables, as in this sentence: “The signature gene lists for the cell clusters and their subpopulations were provided in the spreadsheet files as Tables S1-S3. Authors and the editorial office should be sure readers have access to these supplementary tables.
Author Response
Response to Reviewer 2
Overall, this is an excellent study and should be published with no modifications or only minor modifications. The methods and datasets are very well explained. And, there is an SOM available for data transparency.
Response: We sincerely thank reviewer for the encouragement and suggestions. We have added the suggested information and modifications accordingly.
This reviewer has only three minor requests: An IACUC approval number should be provided, as a matter of the paper trail for anyone interested in verifications; and the following sentence can be changed to “We first conducted a cross-dataset analysis…”; and because of the neutrophil results, addressed at end of Discussion, it would be useful to have some comments about bacteria in the microenvironment.
Response: Thank you for these important comments. We apologize for not including the IACUC approval number in the previous version of this manuscript. We have added the information for the IACUC and IRB approval in the related sections. We have corrected the indicated sentence to “We first conducted a cross-dataset analysis…” as suggested. In this revised manuscript, we also added more discussions on the important topic of bacteria/microbiome in the Discussion section.
There is one final issue: While this reviewer did have access to two supplementary figures, despite repeated requests to the editorial office, this reviewer did not have access to the supplementary tables, as in this sentence: “The signature gene lists for the cell clusters and their subpopulations were provided in the spreadsheet files as Tables S1-S3. Authors and the editorial office should be sure readers have access to these supplementary tables.
Response: Thank you for this important request. We apologize for not having the supplementary tables accessible during the review process for the previous version of this manuscript. We will make sure that the Supplementary Tables 1-3 are accessible for this revised manuscript (as well as the future version of this study).
Reviewer 3 Report
Yang et al leveraged publicly available data sets of single-cell RNA-sequencing (scRNA-seq) from KPC mice. The authors compared sequencing from early primary disease, late primary disease, or liver metastases. These analyses uncovered an interesting shift from classical subtype tumor cells early in disease, to basal-like subtype late in disease, and basal-like with higher expression of a mesenchymal gene signature in metastatic tumor cells. Further, the authors identify a liver metastasis-specific endothelial cell population but a pronounced loss of CAFs in liver mets. Analysis of CAFs from primary tumors revealed loss of aCAFs and bCAFs (resembling iCAFs and a subpopulation of myCAFs, respectively) and gain of c/d/eCAFs (resembling additional myCAFs and apCAFs). Finally, the authors demonstrate loss of T, B, and NK cells as early disease progresses to late disease, and an unexpected shift in myeloid populations. By incorporating scRNA-seq data from bone marrow, the authors conclude that granulocyte-myeloid progenitors present in the bone marrow are also present in liver metastases, suggesting they may give rise to immune populations present in the mets.
The analyses presented here seem rational and provide a clear comparison of cell populations through PDAC progression. Minor concerns are listed below:
Please address:
· Clarify what was considered “early”. Did these samples include PanIN-1 and 2? Or also PanIN-3 and PDAC?
· aCAF and bCAF appear to cluster together, although authors claim they resemble separate CAF populations (iCAFs and myCAFs, respectively). Please show iCAF vs myCAF cell score overlayed on a/b/c/dCAF populations.
· Please include a statement that CAFs are often lost during tissue dissociation for scRNA-seq. It is well documented that CAFs are a major component of liver mets. The text currently suggests that CAFs are depleted in liver mets.
· Line 377-378; the authors state a communal GMP cell population in primary tumors, mets, and healthy bone marrow. However, the data doesn’t support these populations overlaying. Please provide a UMAP plot of only GMP cells from each dataset. The conclusion in lines 380-382 (bone-marrow derived GMPs give rise to some immune populations in tumors) is not supported and should be changed.
Authors may consider addressing:
· Please consider overlaying basal and classical cell signatures on CC-1, CC2, CC-3
· How does EC-2 compare to endothelial cell signature from HCC or liver mets from a different tumor origin?
Minor notes:
· Mice are C57Bl/6 (not C57/BL6)
· Line 367, Npg is repeated
Author Response
Reviewer 3
Yang et al leveraged publicly available data sets of single-cell RNA-sequencing (scRNA-seq) from KPC mice. The authors compared sequencing from early primary disease, late primary disease, or liver metastases. These analyses uncovered an interesting shift from classical subtype tumor cells early in disease, to basal-like subtype late in disease, and basal-like with higher expression of a mesenchymal gene signature in metastatic tumor cells. Further, the authors identify a liver metastasis-specific endothelial cell population but a pronounced loss of CAFs in liver mets. Analysis of CAFs from primary tumors revealed loss of aCAFs and bCAFs (resembling iCAFs and a subpopulation of myCAFs, respectively) and gain of c/d/eCAFs (resembling additional myCAFs and apCAFs). Finally, the authors demonstrate loss of T, B, and NK cells as early disease progresses to late disease, and an unexpected shift in myeloid populations. By incorporating scRNA-seq data from bone marrow, the authors conclude that granulocyte-myeloid progenitors present in the bone marrow are also present in liver metastases, suggesting they may give rise to immune populations present in the mets.
The analyses presented here seem rational and provide a clear comparison of cell populations through PDAC progression. Minor concerns are listed below:
Response: We greatly appreciate the reviewer’s overall enthusiasm and important comments. Accordingly, we have addressed these points with additional analyses and new experiments. We have also added more discussions in this revised manuscript.
Please address:
- Clarify what was considered “early”. Did these samples include PanIN-1 and 2? Or also PanIN-3 and PDAC?
Response: We apologize for not including related information for defining the tumor stages in our previous version of manuscript. We have now added the description in the Methods section to clarify this important point. Early-stage PDAC samples were defined as those harboring less than 10% pancreatic adenocarcinoma areas. Late-stage PDAC samples were defined as those harboring greater than 50% pancreatic adenocarcinoma areas. Therefore, the early-stage PDAC samples defined in this study indeed have PanIN-3 and PDAC, in addition to PanIN-1 and PanIN-2. This definition was previously described in the recently published datasets (GSE198815) by our original study (McAndrews, Chen, Darpolor et al., Cancer Discovery, 2022).
- aCAF and bCAF appear to cluster together, although authors claim they resemble separate CAF populations (iCAFs and myCAFs, respectively). Please show iCAF vs myCAF cell score overlayed on a/b/c/dCAF populations.
Response: We appreciate this important comment. We completely agree with the reviewer that aCAF and bCAF (as adjacent CAF subclusters) share more phenotypical similarities than divergences. The aCAF and bCAF shall be classified as iCAF. According to this comment, we added new gene expression heatmap plot and dot plot overlaying the myCAF/iCAF/apCAF signature genes on a-fCAF subclusters in the new Supplementary Figure 4A and 4B. Based on these new analyses, we have improved our result statements to describe the CAF subpopulations more accurately. Our new results support this comment by this reviewer regarding the phenotypic similarity of aCAF and bCAF.
- Please include a statement that CAFs are often lost during tissue dissociation for scRNA-seq. It is well documented that CAFs are a major component of liver mets. The text currently suggests that CAFs are depleted in liver mets.
Response: Thank you. According to this comment, we have modified our statement and result description.
- Line 377-378; the authors state a communal GMP cell population in primary tumors, mets, and healthy bone marrow. However, the data doesn’t support these populations overlaying. Please provide a UMAP plot of only GMP cells from each dataset. The conclusion in lines 380-382 (bone-marrow derived GMPs give rise to some immune populations in tumors) is not supported and should be changed.
Response: We appreciate this important comment. Accordingly, we added the UMAP plot for the GMP cell cluster alone, across the indicated groups (new Supplementary Figure 5A). We agree with the reviewer that the GMP subcluster between primary tumors and liver metastases reveals slightly different distributions, which is understandable because of the GMP phenotypic shift caused by completely different microenvironments between primary tumors and liver metastases. However, based on our previous results and new analyses, we believe that the possibility of a common GMP cluster is still supported by the specific and robust expression of many essential GMP marker genes including Ramp1, Elane, Mpo, and Ctsg (shown in new Supplementary Figure 5B and previous Figure 8C), which is consistent with recent studies (Alshetaiwi et al., Science Immunology, 2020; Dong et al., Nature Cell Biology, 2020; Giladi et al., Nature Cell Biology, 2018; Veglia et al., Nature Immunology, 2018).
Authors may consider addressing:
- Please consider overlaying basal and classical cell signatures on CC-1, CC2, CC-3
Response: Thank you for this comment. We previously presented the gene expression profiles of basal and classical cell type for CC-1, CC-2, and CC-3 as dot plot in Figure 2E. According to this suggestion, we added new gene expression heatmap plot overlaying the basal and classical cell gene signatures on CC-1, CC2, and CC-3 in the new Supplementary Figure 2B. We observed consistent results showing different basal and classical subtype signatures among CC-1, CC-2, and CC-3 subclusters.
- How does EC-2 compare to endothelial cell signature from HCC or liver mets from a different tumor origin?
Response: We appreciate this intriguing point. We completely agree with the reviewer that it would be interesting to investigate whether the gene signature of EC-2 (identified in the liver metastasis) is comparable to that of endothelial cells in HCC or the liver metastasis of other cancer types. According to this comment, we have conducted a thorough search of current literatures with available scRNA-seq datasets. There are many published papers that investigated the HCC microenvironment using scRNA-seq technique. However, these papers did not thoroughly present the gene signatures of endothelial cell population. In addition, the available scRNA-seq dataset for liver metastases of pancreatic cancer or other cancer types is still lacking. Therefore, we do not have sufficient dataset to investigate this point at the current stage. Nevertheless, we added more discussions on the important point in the Discussion section. We will investigate this point in our future studies if there are more available datasets.
Minor notes:
- Mice are C57Bl/6 (not C57/BL6)
Response: We apologize for this mistake. We have corrected the mouse nomenclature in this revised manuscript.
- Line 367, Npg is repeated
Response: We apologize for this error. We have corrected this in the revised manuscript.
Round 2
Reviewer 1 Report
The revision work is greatly appreciated.